# Preparation and Properties of Acetoacetic Ester-Terminated Polyether Pre-Synthesis Modified Phenolic Foam

**DOI:** 10.3390/ma12030334

**Published:** 2019-01-22

**Authors:** Tiejun Ge, Kaihong Tang, Xiaojun Tang

**Affiliations:** 1Plastic Engineering Research Center of Shenyang University of Chemical Technology, Shenyang 110142, China; t_angkh@163.com (K.T.); tangxj2008@yeah.net (X.T.); 2Liaoning Polymer Materials Engineering and Technology Research Center, Shenyang 110142, China; 3Shenyang Huada and Kangping Plastic Woven Research Institute, Shenyang 110142, China; 4Yingkou Sounrun New Material Engineering Technology Co., Ltd., Yingkou 115000, China

**Keywords:** acetoacetic ester-terminated polyether, pre-synthesis modification, phenolic foam, toughness

## Abstract

In the present study, acetoacetic ester-terminated polyether was selected as a modifier to prepare a new type of polyether phenolic resin, which was successfully prepared by pre-synthesis modification. It is used to prepare interpenetrating cross-linked network structure modified phenolic foam with excellent mechanical properties. Fourier transform infrared spectroscopy (FT-IR) and nuclear magnetic resonance (^1^H NMR, ^13^C NMR) were used to characterize the molecular structure of the polyether phenolic resin. The results showed that the acetoacetic ester-terminated polyether successfully modified the phenolic resin and introduced a polyether skeleton into the resin structure. The effect of changing the added amount of acetoacetic ester-terminated polyether from 10% to 20% of the phenol content on the mechanical properties and microstructure of the modified phenolic foam was investigated. The results showed that when the amount of acetoacetic ester-terminated polyether was 16% the amount of phenol, this resulted in the best toughness of the modified foam, which had a bending deflection that could be increased to more than three times that of the base phenolic foam. The modified phenolic foam cell diameter was reduced by 36.3%, and the distribution was more uniform, which formed a denser network structure than that of the base phenolic foam. The bending strength was increased by 0.85 MPa, and the pulverization rate was as low as 1.3%.

## 1. Introduction

With the rapid development of the construction industry, phenolic foam is widely used in building materials for its excellent flame retardancy [1]. This situation has greatly promoted the improvement of the production process of phenolic foam. Phenolic foam has advantages that other foams do not have, such as flame retardant, self-extinguishing, low toxicity, low smoke, etc. However, its shortcomings are also notable, including its low elongation, brittleness, poor toughness, etc. [2]. The benzene ring in the molecular structure is only connected by a methylene group, and the resin forms a uniform and stable three-dimensional network structure. This structure causes the density of the rigid group (benzene ring) to be large. The packing is tight, the steric hindrance is large, and the degree of freedom of the link rotation is small, resulting in the low elongation and high brittleness of the pure phenolic foam [3,4]. 

Many researchers have modified the shortcomings of phenolic foam, such as using bio-oil as a renewable toughening agent for PF, partially replacing phenol for the preparation of phenolic foam, and reducing the brittleness of phenolic foam [1]. Applying DOPO-ITA modified ethyl cellulose to the preparation of phenolic foam significantly improved the mechanical strength and heat resistance of the modified foam [5]. A modification of a phenolic resin with epoxy methacrylate functionalized silica sols to improve the ablation resistance of their glass fiber-reinforced composites [6]. Mirski et al., who used esters of different carbon chain lengths to modify the phenolic resin for the molecular structure, found that the modified phenolic resin exhibited a shorter gel time and lower activation energy at 130 °C [7]. Through studying the effects of changes in the chemical composition and processing conditions of phenolic foams, it is concluded that the compressive strength and density increase with the increasing acid catalyst concentration, etc. [8]. During the study, dicyandiamide was used as a toughening agent to change the brittleness of phenolic foam, and the compressive strength and impact strength of the modified foam were significantly improved [9].

It can be seen from the research reports on phenolic foam in recent years that the modification of the molecular structure of phenolic resin has gradually become the main research direction for optimizing the performance of phenolic foam [10,11,12]. Aiming at the weak link in the molecular structure of the foamable phenolic resin, the targeted improvement of its performance has become the preferred means of modifying the phenolic foam. Therefore, we selected the pre-synthesis modification method in the chemical modification to improve the performance of phenolic foam. The pre-synthesis modification method [2] first involves the modification of phenol, and then the reaction of the modified phenol with formaldehyde to synthesize a new resin. The synthesized new resin and the base phenolic resin are foam-cured at a certain ratio to obtain a modified phenolic foam with excellent properties.

In recent years, the polyether skeleton with its low viscosity and good solubility has been widely used in materials. Since the ether bond has low cohesive energy and is easy to rotate, the material prepared by it has excellent mechanical properties [13]. Some researchers use polyether to modify materials, such as Yang H. et al., who synthesized two new polyethers and used them as toughening agents to physically modify the phenolic foam, finding that the total heat release rate of the modified foam can be reduced by up to 42% [14]. Some research results show that the cell structure and compression properties of phenolic foams will be different after modification by polyether polyols with different hydroxyl values [15]. Flame-retardant polyether polyols can be used to prepare flame-retardant polyurethane prepolymers for toughening phenolic foams [16], and the evening primrose oil-based polyol can use to modify polyurethane–polyisocyanurate foams with excellent performance and green environmental protection [17]. Zhu et al. synthesized a novel polyetherimide material with excellent mechanical properties [18]. Both melamine and phenolic synthetic polyether polyols and the tar-based Mannich polyether polyol can be used to modify polyurethane foams [19,20,21]. 

As early as 1968, a United States (US) patent disclosed a process for synthesizing a thermoplastic polyhydroxyether having a degree of polymerization of at least 30 using phenyl ether or naphthyl ether as material [22]. It is applied to modified phenolic resins to improve their toughness and impact strength. In 1984, Komatsubara et al. synthesized diphenyl ether modified phenolic resin by reacting novolac with diphenyl ether under the action of an acid catalyst [23]. The consumption rate of diphenyl ether under different catalysts was studied, and it was proved that the curing time of the modified resin increased with the increase of diphenyl ether content. The processes of the above two modification methods and the modification of the novolac-type phenolic resin with the polyhydroxyl ether of bisphenol A [24] and the modification of phenolic resins by aralkyl ethers [25] or anisole [26] are complicated, and the reaction temperature is too high because of the presence of the phenyl ether. Although some researchers have reported the use of polyether skeletons in materials such as sealants, polyurethanes, epoxies [27], and phenolic foams [28], the polyether skeleton has not been incorporated into the molecular structure of the phenolic resin. It is used as a toughening agent to physically modify the phenolic resin [29], and the phenolic foam is obtained by blending and foaming. Therefore, this study chose to introduce the polyether skeleton into the phenolic resin structure and explore the properties of the modified phenolic foam from the perspective of the molecular structure. 

Ge et al. analyzed the effect of reactive polyethers with different molecular weights on the properties of phenolic foams, in which the modified phenolic resin and foam using the reactive polyether with a molecular weight of 1000 showed the best performance [30]. Therefore, we modified the phenolic resin by using an acetoacetic ester-terminated polyether with a molecular weight of about 1000, and cross-linked the obtained polyether resin with the base resin to prepare a modified phenolic foam. The synthetic procedure for preparing acetoacetic ester-terminated polyether is involved in the patent that was published by Groegler in 1969, and is still applied today [31,32]. The polyether skeleton has low cohesive energy and easily rotates, which makes up for the disadvantage that the benzene ring is only connected by methylene groups in the phenolic resin structure and the density of the rigid group is too large to rotate, and uses its unique flexible structure to toughen the phenolic foam. In the paper, the effects of the acetoacetic ester-terminated polyether on the properties (such as the resin molecular structure, foam strength, toughness, pulverization rate, and microstructure) of the phenolic foam were investigated.

## 2. Materials and Methods

### 2.1. Materials

Phenol, paraformaldehyde, sodium hydroxide (as a catalyst), n-pentane (as a blowing agent), and ethyl acetoacetate were purchased from the Tianjin Damao Chemical Reagent Factory (Tianjin, China). The above-mentioned raw materials and reagents were all analytical reagents (≥99.7%). Tween-80 (chemically pure, as a surfactant), hydrochloric acid (guaranteed reagent, as a catalyst), and 98% sulfuric acid (as a curing agent) were supplied by Sinopharm Chemical Reagent Co., Ltd. (Shanghai, China). Polyether diols (1000D) were supplied by Guangzhou Desson Chemical Co., Ltd. (Guangzhou, China). Ethyl acetoacetate and polyether diols were used as raw materials for the synthesis of the acetoacetic ester-terminated polyether.

### 2.2. Synthesis of Modified Phenolic Resin

The acetoacetic ester-terminated polyether synthesis formula includes the ethyl acetoacetate to polyether diols mass ratio of 2:1, and the sulfuric acid catalyst is 0.3% of the total reactant mass.

According to the formula, the ethyl acetoacetate and polyether diols were added to the three-necked flask (Tianjin Damao Chemical Reagent Factory, Tianjin, China) containing a glass manifold (Tianjin Damao Chemical Reagent Factory, Tianjin, China), and the upper part was connected with a reflux condenser. The mixture was placed in a magnetic stirring oil bath at 120 °C. The sulfuric acid catalyst was added after the temperature stabilized and the mixture was allowed to react continuously for about three hours. As shown in Scheme 1, during the reaction of ethyl acetoacetate with the polyether diols, the polyether diols hydroxyl group were removed, so that only the polyether skeleton and the acetoacetate remained in the final product. The cooled liquid was poured out for filtration to obtain the acetoacetic ester-terminated polyether.

According to the formula of Table 1, acetoacetic ester-terminated polyether and phenol were added to the three-necked flask, which was stirred using a paddle (the temperature of the water bath is about 80 °C). Then, a certain amount of hydrochloric acid was added as the catalyst, and evenly stirred for about 30 min. After the liquid in the three-necked flask was cooled to 65 °C, the sodium hydroxide catalyst was added and stirred uniformly. The paraformaldehyde was added into the flask in five batches within half an hour, the temperature of the water bath was maintained for one to two hours, and the paraformaldehyde was left to fully react. The temperature of the water bath was raised to 90 °C, and the mixture reacted continuously for about 30 min. At this time, the resin polymerization reaction was not complete. In order to balance the reaction equation, the cooled resin was needed to seal for 24 h in order to obtain the polyether phenolic resin.

Acetoacetic ester-terminated polyether and phenol were undergoing the electrophilic addition reaction [33] to form an aralkyl ether compound with two benzene rings under the acidic conditions. At this time, the paraformaldehyde was added, and a low cross-link density polyether phenolic resin such as the one shown in Scheme 2 was synthesized under the alkaline conditions.

### 2.3. Preparation of Phenolic Foam

The phenolic resin foaming formulation is shown in Table 2. First, the resin was uniformly mixed according to the formulation shown in Table 2; then, it was mechanically stirred for one minute by adding a surfactant before adding a foaming agent for thorough mixing. A certain amount of curing agent was added and quickly mixed well; then, the resin was poured into the mold (open glass mold) and placed in the oven (Beijing ever bright medical treatment instrument co., Ltd., Beijing, China) (the temperature of the oven is about 70–75 °C) for 10 min to allow it to be cured.

## 3. Characterization

The structure of the phenolic resin and acetoacetic ester-terminated polyether were characterized by Fourier transform infrared spectroscopy (FT-IR) (NEXUS 470 Thermo Electron Corporation, Shanghai, China) and a nuclear magnetic resonance (NMR) spectrometer (AVANCE Ⅲ Bruker Biospin, Munich, Germany). FT-IR tests were performed by directly applying the sample to a pressing potassium bromide troche. NMR tests were performed using DMSO solvent.

The bending performance was determined according to the Chinese National Standard (GB/T 8812-2007), and the compressive strength was determined according to the Chinese National Standard (GB/T 8813-2008) using an RGL-type microcomputer control electronic universal testing machine (Shenzhen Rui Geer Instrument Co., Ltd, Shenzhen, China). The parameters are as follows: support span of 60 ± 1 mm; indenter arc radius of 5 ± 0.2 mm; sample size length of 100 ± 0.5 mm, width of 10 ± 0.5 mm, and thickness of 4 ± 0.2 mm. The number of samples in each group is five, while the head speed is three mm/min. The bending test records the strength and fracture displacement of the foam. The sample is compressed at a rate of 10% of the initial thickness of the compressed sample per minute until the thickness of the sample becomes 85% of the initial thickness. 

The pulverization rate was determined according to the Chinese National Standard (GB/T 3960-2016). The foam water absorption rate was determined according to the Chinese National Standard (GB/T 8810-2005). Contrasting the quality before and after soaking at the conditions of the sample (the environment temperature was 23 ± 2 °C, the relative humidity was 50 ± 5%, and the soak time was about 96 ± one hour). The cell structure was observed with a scanning electron microscope (SEM) (EVO10 Carl Zeiss, Oberkochen, Germany). The cell diameter and diameter distribution of the foam were analyzed using image analysis software (Image-pro plus 6.0, Media Cybernetics, Maryland, America).

## 4. Results and Discussion

### 4.1. Polyether Phenolic Resin Structure

#### 4.1.1. FT-IR Analysis

As shown in Figure 1, the acetoacetic ester-terminated polyether in the polyether phenolic resin was added in an amount of 16% of the amount of phenol. In the infrared spectrum of the polyether phenolic resin, the stretching vibration region of the benzene ring skeleton is located at 1595.75–1457.79 cm^−1^. There are absorption peaks at 756.10 cm^−1^ in the ortho-substitution bending vibration region of the benzene ring, and 829.12 cm^−1^ in the para-substitution bending vibration region of the benzene ring. The IR bands of the C–O group of phenol and C–O group of hydroxymethyl appeared at 1153.18 cm^−1^ and 1108.32 cm^−1^ [2]. Compared with the basic phenolic resin, the IR spectra of polyether phenolic resin showed that the stretching vibration peaks of the polyether methylene are located at 2937.54 cm^−1^ and 2860.77 cm^−1^. 

In the infrared spectrum of the acetoacetic ester-terminated polyether, the stretching vibration region of the C=O group and the —COOR group are located at 1743.42 cm^−1^ and 1719.47 cm^−1^. The stretching vibration peaks of the —COOR group of the polyether phenolic resin are located at 1712.94 cm^−1^. The C=O group of the ketone structure in acetoacetic ester-terminated polyether were undergoing a nucleophilic addition reaction under acidic conditions to form an enol-like structure. The characteristic peaks of the C=O group of the modifier (acetoacetic ester-terminated polyether) are located at 1743.42 cm^−1^ in the resin; these peaks are not visible.

#### 4.1.2. H-NMR Analysis

As shown in Figure 2, δ = 6.5–7.4 ppm at position h is the proton peak of hydrogen in the benzene ring, δ = 4.8–5.1 ppm at position f is the proton peak of hydrogen in phenol, δ = 4.2–4.7 ppm at position e is the hydrogen proton peak of the methyl group in the hydroxymethyl group, δ = 3.6–3.8 ppm at position d is the proton peak of the hydrogen in the linking benzene ring –CH_2_–, and δ = 1.9–2.1 ppm at position c is the hydrogen proton peak of hydroxymethyl –OH [2]. Compared with the basic phenolic resin, the ^1^H NMR spectra of polyether phenolic resin showed that the δ = 1.0–1.2 ppm at position a is the hydrogen proton peak of hydrogen in the polyether skeleton, while δ = 1.7–1.8 ppm at position b is the hydrogen proton peak of hydrogen in the terminal methyl group of acetoacetic ester, while δ = 5.8–5.9 ppm at position g is the hydrogen proton peak of hydrogen in the methylene group of acetoacetic ester. The δ = 3.3 ppm at position one is the proton chemical shift of H_2_O involved in NMR, while the δ = 2.5 ppm at position two is the chemical shift of the deuterated solvent DMSO.

#### 4.1.3. C-NMR Analysis

It can be seen from Figure 3 that the δ = 39.3–40.6 ppm at position one is the chemical shift of the deuterated solvent DMSO. The δ = 154–160 ppm are the peaks of the carbon on the benzene ring in the phenolic hydroxyl group, δ = 112–139 ppm are the peaks of the other carbons on the benzene ring, except for the phenolic hydroxyl group, and δ = 53–70 ppm are the peaks of the carbon in the hydroxymethyl group. Compared with the basic phenolic resin, the ^13^C NMR spectra of the polyether phenolic resin showed that the δ = 15–20 ppm is the peak of the carbon in the methyl group of the acetoacetic ester-terminated polyether, while δ = 15.9 ppm is the peak of the carbon in the methyl group of the polyether skeleton, and δ = 19.1 ppm is the peak of the carbon in the terminal methyl group of enol acetoacetic ester. The δ = 70–75 ppm at position b are the peaks of the carbon in the methylene group and methyne group of polyether skeleton, δ = 109–111 ppm at position c is the peak of the carbon in the methylene group of acetoacetic ester, δ = 154–155 ppm at position d is the peak of the carbon in the acetoacetic ester connected to the benzene ring, and δ = 164–165 ppm at position e is the peak of the carbon in the —COOR group of the acetoacetic ester-terminated polyether. There is no characteristic peak of carbon in the C=O group in the carbon spectrum.

It can be seen from the FT-IR, ^1^H NMR, and ^13^C NMR spectra that the polyether phenolic resin has been successfully synthesized.

### 4.2. Mechanism Analysis

The phenolic foam is a foamed polymer formed by volatilizing a non-toxic foaming gas with temperature, uniformly dispersing in a polymer of phenolic resin and curing with an acidic curing agent. The strength of the foam cell wall determines the toughness and strength of the foam. Under the action of the same amount of foaming agent, the better the strength and toughness of the foam cell wall, the smaller the cell expansion, and the thicker the cell wall, the stronger the force that can be withstood. Since a single polyether phenolic resin is a branched network structure, the degree of cross-linking is lower than that of the basic phenolic resin, and the density of the molecular structure is not sufficient. Therefore, the polyether phenolic resin and the basic phenolic resin are cross-linked and cured at a ratio of 45:55 to form an interpenetrating cross-linked network structure, as shown in Figure 4. In this structure, the polyether phenol resin (A) is interspersed in the network structure of the basic phenol resin (B) and cross-linked with it. The resulting foam structure is more stable, increasing its strength while toughening the phenolic foam. The data regarding the mechanical and other properties can be seen in Appendix A.

### 4.3. The Properties of the Foam

#### 4.3.1. Bending Strength

As the amount of added acetoacetic ester-terminated polyether increases, the bending strength of the sample gradually increases to 0.305 MPa in Figure 5, which was increased to 0.85 MPa compared with the basic phenolic foam. This is because, with the addition of acetoacetic ester-terminated polyether, the space network of foam has gradually changed from an irregular network structure to a dense interpenetrating cross-linking network structure. The intermolecular bond is tighter, and the bonding strength is increased so that the strength of the modified foam is increased [34]. However, when the amount of added acetoacetic ester-terminated polyether is too large, the curing group of the resin is reduced, and the degree of cross-linking is insufficient, resulting in a decrease in the bending strength.

#### 4.3.2. Compressive Strength

As shown in Figure 6, the compressive strength of the modified foam increases first and then decreases as the amount of added acetoacetic ester-terminated polyether increases. The maximum compressive strength is 0.203 MPa when the content of the acetoacetic ester-terminated polyether is 16% of the amount of the phenol. When the amount of acetoacetic ester-terminated polyether added is less than 16% of the amount of the phenol, the compressive strength gradually increases. This is because the increasing addition of polyether acetoacetate caused increases in the intermolecular bond strength, the toughness of the foam, and the maximum load that the foam wall could withstand. Thus, the modified phenolic foam has a higher compressive strength [35,36]. However, when the amount of added acetoacetic ester-terminated polyether exceeds 16%, the space network density of the modified phenolic resin is reduced and the arrangement is irregular, which ultimately decreases the compressive strength of the modified foam.

#### 4.3.3. Fracture Displacement

While testing the bending strength, we also measured the fracture displacement of the sample. Using the fracture displacement to indicate the bending deflection of the sample under the same conditions, the toughness of the phenolic foam was characterized. The greater the fracture displacement of the sample, the better the toughness of the foam.

As shown in Figure 7, the acetoacetic ester-terminated polyether modified phenolic foam has a significant increase in fracture displacement. This is because a polyether skeleton as an alkyl flexible chain is introduced into the phenol resin structure. The polyether skeleton has low cohesive energy and easily rotates, which makes up for the disadvantage that the benzene ring is only connected by methylene groups in the phenolic resin structure and the density of the rigid group is too large to rotate. Thereby, the toughness of the foam is improved, and the flexibility of the modified phenolic foam is increased. When the amount of added acetoacetic ester-terminated polyether was 16%, the bending deflection of the sample was the best, and the fracture displacement was 14.8 mm, which was increased by three times compared to the fracture displacement of the basic phenolic foam. When the added amount exceeded 16%, the bending deflection slightly decreased. When the amount of added acetoacetic ester-terminated polyether increased, the space molecules arrangement of the modified phenolic resin became looser, and the foam strength decreased. Consequently, the brittleness of the foam increased and the toughness decreased.

#### 4.3.4. Pulverization Rate

As shown in Figure 8, the pulverization rate of the modified foam decreased first and then increased as the amount of added acetoacetic ester-terminated polyether increased. The minimum pulverization rate is 1.3% when the content of the acetoacetic ester-terminated polyether is 16% of the amount of the phenol, which decreased by 4.4% compared to the pulverization rate of the basic phenolic foam. This is because the introduction of the polyether skeleton in the resin structure improves the toughness of the modified foam, and the polyether has good wear resistance, which can absorb more force during the rubbing process so that the modified foam cannot be easily destroyed [37,38]. However, when the amount of added acetoacetic ester-terminated polyether is too large, the foam strength and the intermolecular bonding strength are decreased. Consequently, the pulverization rate of the foam increases.

#### 4.3.5. Apparent Density

As shown in Figure 9, the apparent density of the phenolic foam modified by polyether acetoacetate gradually increases. When the amount of polyether acetoacetate exceeds 18%, the foam density exceeds that of ordinary phenolic foam. This is because the strength of the cell wall increases as the amount of acetoacetic ester-terminated polyether added increases. Under the condition that the amount of foaming agent is constant, the gas cannot expand the cell better when it is crushed into the cell [39], so the cell size decreases and results in a greater foam density.

#### 4.3.6. Water Absorption Rate

As shown in Figure 10, the water absorption of the basic phenolic foam is about 8%. The change in the water absorption of the phenolic foam modified by acetoacetic ester-terminated polyether is not noticeable. This is because the addition of acetoacetic ester-terminated polyether increases the toughness of the phenolic foam, improves the toughness of the cell wall, and reduces the bursting of the foaming agent into the cell, thereby decreasing the water absorption of the foam. However, both the ether group and the hydroxyl group in the structure of the acetoacetic ester-terminated polyether are hydrophilic groups [40], so the water absorption of the modified foam is not significantly improved.

#### 4.3.7. Cell Microstructure

As shown in Figure 11, compared with the basic foam, the SEM of the modified foam shows that the cell structure of the foam is three-dimensional, the network structure of the modified foam is denser, and there is almost no gap between the cells. Taking the center point of the normal distribution as the average diameter of the cells [41], it can be seen from the cell diameter distribution figure that the average cell diameter of the modified phenolic foam is reduced by 36.3% compared with the basic phenolic foam. The cell diameter distribution of the modified foam is relatively concentrated, mainly in the range of 60 to 100 μm. Moreover, the number of cells that have a large aperture is extremely small, and the shape of the cells is mostly a uniform polygonal structure. It can be seen from the SEM and cell diameter distribution that the acetoacetic ester-terminated polyether modified phenolic foam forms a denser network structure, the cell diameter is reduced, and cell size and cell wall thickness are more uniform.

## 5. Conclusions

In this paper, acetoacetic ester-terminated polyether was used as a modifier, and the polyether skeleton was successfully inserted into the phenolic resin structure. The polyether phenolic resin and the basic phenolic resin are cross-linked and cured at a ratio of 45:55 to form an interpenetrating cross-linked network structure modified phenolic foam with excellent performance, increasing its strength while toughening the phenolic foam. When the amount of acetoacetic ester-terminated polyether was 16% of the amount of phenol, this resulted in the best toughness of the modified foam, which could be increased more than three times compared to the base phenolic foam. The cell structure shows that the modified phenolic foam has a more regular and denser network structure, the cell diameter was reduced by 36.3%, and the distribution was more uniform, which formed a denser network structure than the base phenolic foam. The bending strength was increased by 0.85 MPa, and the pulverization rate was as low as 1.3%.

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
