# Peer review of "Preparation and Properties of Acetoacetic Ester-Terminated Polyether Pre-Synthesis Modified Phenolic Foam"

_materials, 2019, doi:10.3390/ma12030334_

Round 1

Reviewer 1 Report

In this paper, the authors present an original way to improve the mechanical properties of phenol formaldehyde resin and foam by using reactive polypropylene glycol (PPG) in the stage of resin synthesis. For this, the terminal hydroxyl groups of commercial PPG were transformed to acetoacetic and applied as functional modifier in the resin synthesis. The structure and properties of the obtained materials are comprehensively studied by using standard methods and evaluated in comparison to the basic resin prepared without modifier.

The topic is significant considering the new phenol formaldehyde material developed. The originality of the work is in the acetoacetic ester-terminated PPG used as modifier. The results presented in the paper could be of interest to the research community. However schematic presentation of chemical structures of PPG modifier (page 3; Scheme 1) and of the resin (page 11; Scheme 2) are not convincing as well as the discussion on page 11 (3.3. Mechanism Analysis). In addition, the manuscript needs not only overall language improvement but also scientific terms and style corrections (see below).

In my opinion the manuscript needs major revision. Comments and questions to be considered:

1. Some specific scientific terms used in the manuscript are different from the standard terms, e.g. “pre-synthesis modification”, “polyether diol N210”, “sulfuric acid ω=98%”, “acetoacetic ester-terminated polyether synthesis formula”, “glass water separator”, “potassium bromide abrasive sheet”, etc. This makes the text unclear and I would recommend thorough language and style professional editing of the manuscript.  

2. Abstract, line 20:

“The effect of changing the added amount of acetoacetic ester-terminated polyether on the mechanical properties and microstructure of the modified phenolic foam was investigated.”

It would be useful to specify the range of modifier variation (e.g. from 10 to 20% of the phenol content).

3. Introduction, lines 77-90: Although the choice of modifier and synthetic strategy are explained in the last paragraph of the introduction, the aim of the research is not clearly presented. Please add a short paragraph stating the main goal of this investigation and how it will be achieved.

4. Lines 101-110: 2.2. Synthesis of Modified Phenolic Resin

Please state if the synthetic procedure is newly developed or provide reference(s) if it is adopted or modified from published work(s)? Please add information about the reaction yield and specify the isolation and purification procedure.

5. Line 120: Scheme 1 (this applies for Scheme 2 and entire section 3.3. Mechanism Analysis as well)

Chemical structures of acetoacetic ester-terminated PPG on Scheme 1 (as well as of resin on Scheme 2) is not convincing. It is well known that the molecule of ethyl acetoacetate (a β-ketoester) undergo dynamic process of intramolecular hydrogen bonding, i.e. keto-enol tautomeria is present:

(see the scheme from Thermo Fisher Scientific Inc. Application note AN52327_E 12/17M in the attached file)

 The equilibrium is shifted to the keto-form and the content of enol isomer is estimated by different authors to be about 10% only. It is worth to mention that numerous research papers report on the keto-enol tautomerism of ethyl acetoacetate studied by different methods including NMR, FTIR and UV spectroscopy. It is misleading to present the keto and enol isomers of equal importance in studied reactions. In this context the chemical structures presentation and overall discussion should be reconsidered. If the authors have specific quantitative data on the keto-enol composition of the synthesized acetoacetic ester-terminated PPG it is worth to include it in the manuscript and discuss it in details.

6. Line 121:

“The polyether phenolic resin is obtained by sealing the cooled resin for 24 hours.”

Unclear sentence - please rephrase.

7. Line 122: Table 1 and further: Please specify the meaning of (phr)

8. Lines 158-160:

“The stretching vibration peaks of the —COOR group of the acetoacetic ester- terminated polyether are located at 1712.94 cm-1. Moreover, the characteristic peaks of the C=O group of the modifier (acetoacetic ester-terminated polyether) are located at 1743.42 cm-1 which in the resin are disappeared.”

The discussion on the FTIR spectra should be enhanced. As the C=O band is double, please specify the wavenumber of the second peak too. Please explain the reason why the peak at 1743.42 cm-1 is disappearing.

9. Lines 164 and 179: Figures 2 and 3: The signals in the NMR spectra are marked with labels and numbers and further assigned in the text. It is advisable to include chemical structures with assigned labels and numbers on the spectra for clarity.

10. Line 393: Reference 17

“17. Joanna P. S.; Marcin B.; Boguslaw C.; Marek I. Effect…”

Please check the authors’ family names.

Author Response

Dear Reviewer:

Thank you for your comments concerning our manuscript entitled “Preparation and properties of acetoacetic ester-terminated polyether pre-synthesis modified phenolic foam”. Those comments are very helpful for revising and improving our paper, as well as the important guiding significance to our researches. We have studied comments carefully and try our best to improve the manuscript which we hope meet with approval. Revised portion are marked in red in the paper. The main corrections and the responds are as flowing:

Point 1: Some specific scientific terms used in the manuscript are different from the standard terms, e.g. “pre-synthesis modification”, “polyether diol N210”, “sulfuric acid ω=98%”, “acetoacetic ester-terminated polyether synthesis formula”, “glass water separator”, “potassium bromide abrasive sheet”, etc. This makes the text unclear and I would recommend thorough language and style professional editing of the manuscript. 

Response 1: We have already polished the language by the MDPI platform when we submitted the manuscript. Considering your comments, we contacted the English editor of MDPI again and revised the manuscript. We have revised “polyether diol N-210”,98% sulfuric acid”,  “glass manifold”, “pressing potassium bromide troche” , etc.  But sorry that we unable edit “pre-synthesis modification” and “acetoacetic ester-terminated polyether synthesis formula”.

The “pre-synthesis modification” is to name the procedure of the modified phenol reacted with paraformaldehyde to synthesize the phenolic resin which proposed after we summarize the synthesis method of phenolic resin. During the reaction of ethyl acetoacetate with the polyether diol, the polyether diol hydroxyl group are removed, and only remained the polyether skeleton and the acetoacetate in the final product. So it called “acetoacetic ester-terminated polyether synthesis formula”.

If you have any better suggestions, please let us know. Thank you very much.

Point 2: Abstract, line 20: “The effect of changing the added amount of acetoacetic ester-terminated polyether on the mechanical properties and microstructure of the modified phenolic foam was investigated.”

It would be useful to specify the range of modifier variation (e.g. from 10 to 20% of the phenol content).

Response 2: Thank you and we have added the range of modifier variation to the Abstract.

Point 3: Introduction, lines 77-90: Although the choice of modifier and synthetic strategy are explained in the last paragraph of the introduction, the aim of the research is not clearly presented. Please add a short paragraph stating the main goal of this investigation and how it will be achieved.

Response 3: We have added the main goal of the investigation and the methods to the Introduction.

Point 4: Lines 101-110: 2.2. Synthesis of Modified Phenolic Resin

Please state if the synthetic procedure is newly developed or provide reference(s) if it is adopted or modified from published work(s)? Please add information about the reaction yield and specify the isolation and purification procedure.

Response 4: The synthetic procedure for preparing acetoacetic ester-terminated polyether is involved in the patent published by Groegler G. in 1969 and still applied today [1-2]. The synthetic procedure of the polyether phenolic resin by reacting the modified phenol reacted with paraformaldehyde referred to the synthetic procedure of boron phenolic resin [3] and Xylok resin [4]. After the research group investigated and summarized the phenolic resin modification method, we named the method of the modified phenol reacted with paraformaldehyde to synthesize the phenolic resin as pre-synthesis modification method.

The final product phenolic foam needs to be measured for free formaldehyde emission less than 1.5mg/L and it can be applied after reaching the standard according to the Chinese National Standard GB/T 20974-2014. Since the free formaldehyde in a certain content range does not need to be removed, we did not calculate the yield of the resin. If the resin needs to calculate the yield, the formula is as follows.

This paper mainly studies the properties of modified phenolic foams, in which acetoacetic ester-terminated polyether was selected as a modifier. The carbonyl of the keto-structure in the acetoacetic ester-terminated polyether was undergoing nucleophilic addition reaction under acidic conditions [5], forming an enol-like structure, and the final modifier were reacted with phenol in the form of an enol structure during the experiment. Therefore, we believe that the keto and enol isomers of the acetoacetic ester-terminated polyether have no direct effect on the properties of the modified phenolic resin, so we have no separated and purified the modifier. The resin was only subjected to vacuum dewatering treatment during the synthesis procedure. So we unable provide the isolation and purification procedure, hope you can understand.

 1.      Groegler G. Process of the Product ion of Polyamide foams and Elastomers from Amino crotonic Acid Esters. United States Patent Office 1972, US 3691112.

2.      Yu W. The Synthetic Method of Amine-Terminated Polyether and the Application. Polyurethane Industry 2002, 17, 1-5. [10.3969/j.issn.1005-1902.2002.01.001]

3.      Abdalla M.O., Ludwick A., Mitchell T. Boron-modified phenolic resins for high performance applications. Polymer 2003, 44, 7353-7359. [10.1016/j.polymer.2003.09.019]

4.      Yang G., Gao L., Cheng K. Preparation and properties of quartz cloth-reinforced Xylok composites fabricated by vacuum bag only process. High Performance Polymers 2013, 25, 493-501. [10.1177/0954008312471066]

5.      Riant O., Jérôme H. Asymmetric catalysis for the construction of quaternary carbon centres: nucleophilic addition on ketones and ketimines. Cheminform 2007, 5, 873-888. [10.1039/b617746h]

Point 5: Line 120: Scheme 1 (this applies for Scheme 2 and entire section 3.3. Mechanism Analysis as well)

Chemical structures of acetoacetic ester-terminated PPG on Scheme 1 (as well as of resin on Scheme 2) is not convincing. It is well known that the molecule of ethyl acetoacetate (a β-ketoester) undergo dynamic process of intramolecular hydrogen bonding, i.e. keto-enol tautomeria is present:

(see the scheme from Thermo Fisher Scientific Inc. Application note AN52327_E 12/17M in the attached file)

 The equilibrium is shifted to the keto-form and the content of enol isomer is estimated by different authors to be about 10% only. It is worth to mention that numerous research papers report on the keto-enol tautomerism of ethyl acetoacetate studied by different methods including NMR, FTIR and UV spectroscopy. It is misleading to present the keto and enol isomers of equal importance in studied reactions. In this context the chemical structures presentation and overall discussion should be reconsidered. If the authors have specific quantitative data on the keto-enol composition of the synthesized acetoacetic ester-terminated PPG it is worth to include it in the manuscript and discuss it in details.

Response 5: Thank you for your suggestion on the article. We strongly agree with your statement about the molecule of ethyl acetoacetate has keto-enol tautomeria. As you said, the keto and enol isomers of unequal importance in studied reactions. This is a question worthy of further discussion.

But as stated Response 4, the carbonyl of the keto-structure in the acetoacetic ester-terminated polyether was undergoing nucleophilic addition reaction under acidic conditions, forming an enol-like structure, and the final modifier were reacted with phenol in the form of an enol structure during the experiment. The acetoacetic ester-terminated polyether has been used in the polyurethane industry [1,6-8]. In the synthesis of polyether imide materials, the keto and enol content did not directly affect the preparation and properties of the final product. Therefore, we believe that the keto and enol content will not affect the preparation and properties of the polyether phenolic resin, and we have no tested the keto and enol content of acetoacetic ester-terminated polyether in the experiment. So we unable provide the quantitative data on the keto-enol composition of the synthesized acetoacetic ester-terminated polyether.

Although we also want to improve the level of the manuscript, the quantitative analysis of keto and enol requires a large number of experiments, The authors are facing the problem of master's paper, so we are sorry that we do not have enough time to conduct this research. Your suggestions are important to us. Because of your suggestions, we have found some shortcomings in my current work. We will improve the scientific research level according to your suggestions in the future work and get more achievements!

6.      Radionova E.S., Fedorova O.V., Titova Y.A., et al. Synthesis of podands with dihydropyrimidine fragments based on polyethers with terminal acetoacetamide groups. Chemistry of Heterocyclic Compounds 2015, 51, 478-482. [10.1007/s10593-015-1723-4]

7.      Angelov P. Enamine-Based Domino Strategy for C-Acylation/Deacetylation of Acetoacetamides: A Practical Synthesis of β-Keto Amides. Cheminform 2010 41, 38-077. [10.1002/chin.201038077]

8.      Zhu B.; Xu Q.; Wang G.; Hu C. Synthesis and Characterization of Acetoacetic Ester-Terminated Polyether. Journal of Functional Polymers 2009, 22, 332-336. [10.1145/1651587.1651601]

Point 6: Line 121:The polyether phenolic resin is obtained by sealing the cooled resin for 24 hours.”

Unclear sentence - please rephrase.

Response 6: We have rephrased the sentence in the manuscript.

Point 7: Line 122: Table 1 and further: Please specify the meaning of (phr)

Response 7: The “phr” means parts per hundreds in weight of resin.

Point 8: Lines 158-160:“The stretching vibration peaks of the —COOR group of the acetoacetic ester- terminated polyether are located at 1712.94 cm-1. Moreover, the characteristic peaks of the C=O group of the modifier (acetoacetic ester-terminated polyether) are located at 1743.42 cm-1 which in the resin are disappeared.”

The discussion on the FTIR spectra should be enhanced. As the C=O band is double, please specify the wavenumber of the second peak too. Please explain the reason why the peak at 1743.42 cm-1 is disappearing.

Response 8: In the FTIR spectra, we have relabeled the peaks of C=O and COOR and corrected the description of the discussion on the FTIR spectra, the reason of the peak at 1743.42 cm-1 disappearing was explained.

Point 9: Lines 164 and 179: Figures 2 and 3: The signals in the NMR spectra are marked with labels and numbers and further assigned in the text. It is advisable to include chemical structures with assigned labels and numbers on the spectra for clarity.

Response 9: Thank you very much for your valuable comments. We have put the chemical structures with assigned labels and numbers on the spectra.

Point 10: Line 393: Reference 17

17. Joanna P. S.; Marcin B.; Boguslaw C.; Marek I. Effect…”  

Please check the authors’ family names.

Response 10: We are very sorry for our incorrect writing of the authors’ family names of Reference 17, we have revised it.

Special thanks to you for your good comments and suggestions, and hope that the correction will meet your approval.

Reviewer 2 Report

In this paper, a new type of polyether phenolic resin is prepared by modification by acetoacetic ester-terminated polyether. Phenolic resin was successful in introduction of polyether skeleton in the resin structure. The correlation of amount of acetoacetic ester-terminated polyether with the mechanical properties of phenolic resin is developed. The paper is well written and results are described well. The paper could be accepted for publication in “Materials” after addressing te concerns listed below

-          A Table of comparison of different modifiers along with their anticipated and actual achieved effect on the mechanical and other properties would be nice from reader point of view

-          The method should be compared with other ether. These are just examples, other references should also be searched

-          https://doi.org/10.1295/koron.41.97

-          https://patents.google.com/patent/US3409581

-           

-Line 34.  “Phenolic foam has the advantage that other 34 foams do not have, but its shortcomings are also very obvious”

Elaborate the advantages and shortcoming

-Line 113-121, Line 124-129- Excessive use of “We” and “After”. Rephrase, better to write in third person as in lines 102-109

-Table 1 and Table 2- What is “phr”, define clearly

-Figure 2 and 3. It would be nice to have the structure of the compound in the Figure and then show all the assigned peaks in the spectra

Author Response

Dear Reviewer:

Thank you for your comments concerning our manuscript entitled “Preparation and properties of acetoacetic ester-terminated polyether pre-synthesis modified phenolic foam”. Those comments are very helpful for revising and improving our paper, as well as the important guiding significance to our researches. We have studied comments carefully and have made correction which we hope meet with approval. Revised portion are marked in red in the paper. The main corrections and the responds are as flowing:

Point 1: A Table of comparison of different modifiers along with their anticipated and actual achieved effect on the mechanical and other properties would be nice from reader point of view.

Response 1: Thank you for your suggestion on the article. We have supplemented a Table about the mechanical and other properties of phenolic foam in the Supplementary material.

Point 2: The method should be compared with other ether. These are just examples, other references should also be searched

https://doi.org/10.1295/koron.41.97

https://patents.google.com/patent/US3409581

Response 2: We have added a paragraph in the manuscript based on your suggestion.

Point 3: Line 34.  “Phenolic foam has the advantage that other 34 foams do not have, but its shortcomings are also very obvious”

Elaborate the advantages and shortcoming

Response 3: We have added the advantages and shortcomings in the manuscript.

Point 4: Line 113-121, Line 124-129- Excessive use of “We” and “After”. Rephrase, better to write in third person as in lines 102-109

Response 4: We have rephrased these sentences in the manuscript based on your suggestion, and edited the manuscript by English editing of MDPI.

Point 5: Table 1 and Table 2- What is “phr”, define clearly

Response 5: The “phr” means parts per hundreds in weight of resin.

Point 6: Figure 2 and 3. It would be nice to have the structure of the compound in the Figure and then show all the assigned peaks in the spectra

Response 6: Thank you very much for your valuable comments. We have put the chemical structures with assigned labels and numbers on the spectra.

Special thanks to you for your good comments and suggestions, and hope that the correction will meet your approval.

Reviewer 3 Report

The manuscript titled „Preparation and properties of acetoacetic ester-terminated polyether pre-synthesis modified phenolic foam” by Tiejun Ge et al. contains interesting results.

In my opinion, the Authors could re-edit lines 40-52 and 66-76. Each sentence begins with a surname, which makes the Introduction section unclear. The Characterization section should be divided into individual studies, which should be described in detail.

Author Response

Dear Reviewer:

Thank you for your comments concerning our manuscript entitled “Preparation and properties of acetoacetic ester-terminated polyether pre-synthesis modified phenolic foam”. Those comments are very helpful for revising and improving our paper, as well as the important guiding significance to our researches. We have studied comments carefully and have made correction which we hope meet with approval. Revised portion are marked in red in the paper. The main corrections and the responds are as flowing:

Point 1: In my opinion, the Authors could re-edit lines 40-52 and 66-76. Each sentence begins with a surname, which makes the Introduction section unclear.

Response 1: Thank you for your suggestion on the manuscript. We have rephrased these sentences in the manuscript by English editing of MDPI.

Point 2: The Characterization section should be divided into individual studies, which should be described in detail.

Response 2: Thank you very much for your valuable comments. We have divided and detail described the Characterization section.

Special thanks to you for your good comments and suggestions, and hope that the correction will meet your approval.

Round 2

Reviewer 1 Report

I appreciate the efforts of the authors to answer the comments and questions raised. However most of the extensive explanations and additional information provided in the authors' response to reviewer's comments are not included in the revised manuscript. I strongly advise the authors to update the manuscript in accordance to this discussion and to include the provided additional references.

In the response to reviewer's comments the authors stated: We have revised “polyether diol N-210”… but in the revised manuscript the chemical structure of polyether diol N-210 is still not defined. The same is valid for the abbreviation phr - its meaning should be clearly defined in the manuscript.  

In the authors’ response the synthetic procedure for preparing acetoacetic ester-modified polyether is thoroughly described. I would recommend including this discussion with corresponding additional references in the manuscript as well.

As to the chemical structures presented on Scheme 1 and 2 as well as in section 4.3. Mechanism Analysis, the authors agree that the keto and enol isomers of ethyl acetoacetate are in dynamic equilibrium but keep the misleading schemes showing a mixture of two individual compounds. This makes the processes difficult to understand. Moreover, in the response it is claimed that “the carbonyl of the keto-structure in the acetoacetic ester-terminated polyether was undergoing nucleophilic addition reaction under acidic conditions, forming an enol-like structure, and the final modifier were reacted with phenol in the form of an enol structure during the experiment” but the schemes as presented dо not contribute to the chemical structure elucidation.

In general, the discussion in section 4.3. Mechanism Analysis is rather speculative and not supported by experimental data. I would recommend omitting this part or considerable shortening and moving it immediately after the section of FTIR and NMR analyses (where the proposed chemical structures could be better explained in correlation to the experimental analyses data). 

Author Response

Dear Reviewer:

Thank you for reviewing our manuscript entitled “Preparation and properties of acetoacetic ester-terminated polyether pre-synthesis modified phenolic foam” again. We would like to express our heartfelt thanks to you for your comments. We have studied these comments carefully and have updated the manuscript base on your suggestion which we hope meet with approval. Revised portion are marked in red in the manuscript. The main corrections and the responds are as flowing:

Point 1: In the response to reviewer's comments the authors stated: We have revised “polyether diol N-210”… but in the revised manuscript the chemical structure of polyether diol N-210 is still not defined. The same is valid for the abbreviation phr - its meaning should be clearly defined in the manuscript.

Response 1: We referred some articles revised the “polyether diols” [1,2] and added the meaning of “phr” [3,4] to the footer of Table 1.

1.      Lan P N., Corneillie S., Schacht E., et al. Synthesis and characterization of segmented polyurethanes based on amphiphilic polyether diols. Biomaterials 1997, 17, 2273-2280. [10.1016/0142-9612(96)00056-7]

2.      Pohl M., Danieli E., Leven M., et al. Dynamics of Polyether Polyols and Polyether Carbonate Polyols. Macromolecules 2016, 49, 8995-9003. [10.1021/acs.macromol.6b01601]

3.      Laskoski M., Dominguez D.D., Keller T.M. Synthesis and properties of a liquid oligomeric cyanate ester resin. Polymer 2006, 47, 3727-3733. [10.1016/j.polymer.2006.03.097]

4.      Chen J., Wang G., Zeng X., et al. Toughening of polypropylene–ethylene copolymer with nanosized CaCO3 and styrene–butadiene–styrene. Journal of Applied Polymer Science 2004, 94, 796-802. [10.1002/app.20925]

Point 2: In the authors’ response the synthetic procedure for preparing acetoacetic ester-modified polyether is thoroughly described. I would recommend including this discussion with corresponding additional references in the manuscript as well.

Response 2: Thank you very much for your comment. We have added the synthetic procedure and the references in the manuscript.

Point 3: As to the chemical structures presented on Scheme 1 and 2 as well as in section 4.3. Mechanism Analysis, the authors agree that the keto and enol isomers of ethyl acetoacetate are in dynamic equilibrium but keep the misleading schemes showing a mixture of two individual compounds. This makes the processes difficult to understand. Moreover, in the response it is claimed that “the carbonyl of the keto-structure in the acetoacetic ester-terminated polyether was undergoing nucleophilic addition reaction under acidic conditions, forming an enol-like structure, and the final modifier were reacted with phenol in the form of an enol structure during the experiment” but the schemes as presented dо not contribute to the chemical structure elucidation.

Response 3: We are very sorry that the schemes have not been revised in the previous response. This is our negligence. Your suggestions are very valuable. Based on your suggestion, we have reanalyzed the FTIR and NMR, revised the schemes in the manuscript. We hope that the revised schemes would show the results more clearly.

Point 4: In general, the discussion in section 4.3. Mechanism Analysis is rather speculative and not supported by experimental data. I would recommend omitting this part or considerable shortening and moving it immediately after the section of FTIR and NMR analyses (where the proposed chemical structures could be better explained in correlation to the experimental analyses data).

Response 4: Based on your suggestion, we have shortened the section 4.3. Mechanism Analysis and have moved it after the section of FTIR and NMR analyses.

Special thanks to you for your reviewing and valuable comments. We hope that the correction will meet your approval. Wish you have a nice day.

Round 3

Reviewer 1 Report

I highly appreciate the willingness of the authors to follow the reviewers' recommendations. Most of the reviewers’ comments and suggestions have been taken into account and the present third version of the manuscript is much clearer and enhanced. I would recommend this paper for publication in Materials.